

# The Early Childhood Oral Health Impact Scale (ECOHIS): psychometric properties and application on preschoolers

Bianca Núbia Souza Silva[1], Lucas A. Campos[1,2,3,4], João Marôco[5] and Juliana A.D.B Campos[6]

[1] Department of Morphology and children's clinics, São Paulo State University, Araraquara, São Paulo, Brazil
[2] Faculty of Medicine and Health Technology, Tampere, Finland
[3] Department of Ear and Oral Diseases, Tampere University Hospital, Tampere, Finland
[4] Faculty of Health Sciences, University of Eastern Finland, Kuopio, Finland
[5] William James Center for Research, University Institute of Psychological, Social, and Life Sciences, Lisboa, Portugal
[6] School of Pharmaceutical Sciences, São Paulo State University, Araraquara, São Paulo, Brazil

## ABSTRACT

**Background.** The concept of oral health related to quality of life involves the impact that oral health has on an individual's well-being. The Early Childhood Oral Health Impact Scale (ECOHIS) was developed to measure the impact of oral health problems on the lives of children and their families.

**Objective.** To evaluate the psychometric properties of ECOHIS applied to mothers of preschool children and estimate the influence of demographic characteristics, caries experience, and plaque index on the ECOHIS score.

**Methods.** The fit of ECOHIS to the data was assessed by confirmatory analysis. Chi-square for degrees of freedom ratio ($\chi^2$/df), Comparative Fit Index (CFI), Tucker-Lewis Index (TLI), and Root Mean Square Error of Approximation (RMSEA) were used. Reliability was estimated by the ordinal coefficients alpha ($\alpha$) and omega ($\omega$). The factorial invariance was estimated by the difference in CFI ($\Delta$CFI). Comparisons of the ECOHIS mean scores according to the demographic characteristics, caries experience, and plaque index was performed using analysis of variance (ANOVA).

**Results.** A total of 371 children participated in the study. Mothers' mean age was 33.0 (SD = 7.04) years. The ECOHIS presented a good fit to the data ($\chi^2$/df = 4.31; CFI = 0.95; TLI = 0.94; RMSEA = 0.09) and a strict model invariance. Children without caries and from higher income class had lower oral health impact.

**Conclusion.** The data obtained with the ECOHIS were valid, reliable, and invariant. Children with caries experience and from lower income families had a greater impact of oral problems.

Corresponding author
Lucas A. Campos,
lucas.arraisdecampos@tuni.fi

# INTRODUCTION

Quality of life can be defined as the self-perception of general well-being, and is influenced by culture, principles, objectives, expectations, paradigms, and concerns (*WHO, 1998*). The oral health-related quality of life is a concept related to self-perceived impact of oral

health or oral disease on general well-being (*Baker, 2007*) and is a component of general quality of life, which has been extensively studied (*Chaffee et al., 2017*). Different theoretical approaches have been proposed to assess oral health-related quality of life (*Antunes et al., 2020*; *Barasuol et al., 2020*).

Oral diseases can negatively affect children's well-being, in addition to being painful and affecting growth, socialization, self-esteem, learning, and behavior (*Antunes, Leao & Maia, 2012*; *Bönecker et al., 2012*). Despite advances in understanding the importance of oral health for general well-being, oral health is usually measured quantitatively (*Locker & Allen, 2007*) based on the presence or absence of disease and its severity and consequences. The impact of such conditions on patients and their family is seldom considered (*Locker & Allen, 2007*). Thus, in the last decades, oral health measures have incorporated the assessment of the psychological, physical, and social impact of oral conditions in people's lives (*Sischo & Broder, 2011*), expanding the possibilities for investigation (*Bennadi & Reddy, 2013*).

The earlier the impact of oral diseases is measured, the greater the opportunity to intervene with educational and preventive approaches. Such interventions can be highly effective in school children, as it is during this period that physical and cognitive development occurs and several habits and values are established including those related to health and self-care (*Figueira & Leite, 2008*). However, young children are still cognitively immature, and measuring their oral health impact can be a challenge as they might have difficulties in reporting specific oral health situations, especially with regard to past events (*Rebok et al., 2001*). Thus, as the family members can be directly affected by a child's quality of life and vice-versa (*Pal, 1996*) an alternative for measuring oral health impact on children is obtaining the information through parents or legal guardians (*Filstrup et al., 2003*).

The Early Childhood Oral Health Impact Scale (ECOHIS) is one of the instruments available to assess the impact of oral health problems on preschool children and their families. It was developed by *Pahel, Rozier & Slade (2007)* in English, and later translated into Portuguese by *Tesch, Oliveira & Leão (2008)*. As the identification of oral diseases impact can encourage subsidies for the development of prevention and dental intervention strategies, this study aimed to evaluate the psychometric properties of the ECOHIS when applied to mothers of preschool children and assess the influence of demographic and oral characteristics on the ECOHIS score.

## MATERIALS & METHODS

### Procedures and ethical aspects

The study was based on the ethical principal of the Resolution 466/2012 of the National Health Council. Ethical approval was provided by the Research Ethics Committee of the Faculty of Pharmaceutical Sciences of Araraquara (UNESP) (CAAE 18713419.4.0000.5426) and participants gave their written informed consent. The study design and the reporting of the results were done with the STROBE tool (Strengthening the Reporting of Observational Studies in Epidemiology) (*Von Elm et al., 2007*).

## Study design and sample selection

This was an observational, cross-sectional study. Preschool children (4 to 6 years old) enrolled in public educational institutions in the municipality of Araraquara-SP (Child Education and Recreation Centers-CER), and their mothers participated in the study. The Authorization to carry out the study on the CERs was obtained from the Municipal Education Secretary of the municipality.

The calculation of the minimum sample size was performed using $\beta = 20\%$, $\varepsilon = 12.5\%$, $N = 2,272$ (number of preschool children enrolled in CER) and a 41.8% caries prevalence in 5-year-old children in the State of São Paulo estimated from the Oral Health Project Brazil (*SB Brasil, 2010*). Thus, the minimum estimated sample size was 298. To compensate for a 15% loss rate, the sample was corrected to 351, which also met the demand for statistical analysis for the ECOHIS (27 parameters × 5 subjects per parameter (*Hair et al., 2018*; $n = 135$).

## Sample characteristics

The mothers completed a demographic questionnaire about the child's sex, age and risk factors for caries and mother's age, educational level, and work activity. The economic level of family members was estimated using the Brazil Economic Classification Criterion-ABEP (*Brazilian Market Research Association (ABEP), 2019*).

## Oral clinical examination

A single examiner was previously calibrated in a pilot study with 25 children, and the visible plaque index and the caries index were recorded. The dmft was recorded two times, one week apart, with a high intra-examiner reproducibility (intraclass correlation coefficient = 0.998; 95% CI [0.995−0.999]).

The oral examinations were performed with the children sitting in school chairs under natural light, using wood spatulas and gloves. To determine the bacterial plaque index, the Simplified Oral Hygiene Index (OHI-S) for children was used (*Greene & Vermillion, 1964*; *Pacheco et al., 2013*). The presence of plaque was verified on the vestibular surfaces of the upper deciduous second molars; lingual surfaces of the lower deciduous second molars; vestibular surface of the upper right central incisor; and lingual surface of the lower left central incisor. The index was calculated based on quantitative criteria, for which the tooth surfaces were divided into thirds and evaluated according to scores ranging from 0 to 3 (0: no plaque, 1: less than 1/3 of the tooth surface covered by plaque, 2: from 1/3 to 2/3 of the tooth surface covered by plaque, 3: more than two thirds of the tooth covered by plaque).The final quotation was performed by the sum of the values, divided by the number of teeth examined. Dental caries was diagnosed based on the World Health Organization (WHO) criteria using the dmft (number of decayed, missing due to caries and filled teeth in the primary dentition) (*Organização Mundial da Saúde (OMS), 1999*) the teeth will be examined by quadrants, in the following order 55 to 51; 61 to 65; 75 to 71 and 81 to 85. Dmft was dichotomized considering the absence (dmft = 0) and presence (dmft ≥ 1) of caries.

## Measuring instrument

In the present study, the Portuguese version of the ECOHIS proposed by *Tesch, Oliveira & Leão (2008)* was used. The scale contains thirteen items distributed in two factors. Items 1 to 9 assess the oral problems' impact on the child and items 10 to 13 assess the impact of the child's oral problems on his family. Responses are given in a 5-point Likert type scale. All items must be answered by the child's mother.

## Evaluation of psychometric parameters

The sensitivity of the ECOHIS was assessed by means, medians, and standard deviations and distribution (skewness and kurtosis). The absolute values of kurtosis <7 and skewness <3 indicated no serious deviations from normal distribution (*Marôco, 2014*).

To test the fit of the two-factor structure of the instrument, confirmatory factor analysis (CFA) was performed using the Weighed Least Squares Mean and Variance Adjusted (WLSMV) estimation method. The goodness of fit was tested with the chi-square for degrees of freedom ratio ($\chi^2$/df), the Comparative Fit Index (CFI), the Tucker-Lewis Index (TLI), and the Root Mean Square Error of Approximation (RMSEA) (*Marôco, 2014*; *Byrne, 2010*). The factor loading ($\lambda$) was considered adequate when $\geq 0.50$ and the model was considered to have a good fit when $\chi^2$/df $\leq 5.0$, CFI and TLI $\geq 0.90$, and RMSEA $\leq 0.10$ (*Marôco, 2014*). The modification indices estimated from the Lagrange multiplier (ML) method were also calculated and ML values >11 were inspected.

Convergent validity was assessed based on *Fornell & Larcker (1981)*, who recommended the calculation of the average variance extracted (AVE), considered adequate if $\geq 0.50$. Discriminant validity was estimated using correlational analysis to assess whether items from one factor are not strongly correlated with another factor, and considered adequate when $AVE_i$ and $AVE_j \geq r_{ij}^2$.

The reliability of the ECOHIS was estimated from the ordinal coefficient alpha ($\alpha$) and omega ($\omega$), and satisfactory internal consistency was considered when $\alpha$ and $\omega \geq 0.70$ (*Marôco, 2014*).

The above analyses were performed with the "lavaan" (*Rosseel, 2012*) and "semTools" (*Jorgensen et al., 2019*) packages in the R program (*R core Team, 2022*).

## Factorial invariance

Initially, CFA was performed for each sub-sample (test sample: $n = 194$; validation sample: $n = 177$). Then, the measurement invariance of the factorial model was evaluated using multi-group analysis and CFI difference ($\Delta$CFI). $\Delta$CFI was calculated for the configural and metric models ($\Delta CFI_{M1-M0}$) and for the metric and scalar models ($\Delta CFI_{M2-M1}$). Reduction of up to 0.01 in the CFI indicated measure invariance (*Cheung & Rensvold, 2002*).

## Mean scores of the ECOHIS

The mean scores of the items for the fitted ECOHIS model were calculated. The scores of the following subgroups were compared: sex of the child (male; female), age of the mothers (<30 years; $\geq 30$ years), socioeconomic stratum D–E (mean monthly income: USD$ 175.63); C (USD$ 419.54–735.50); B (USD$ 1,330.09–2,575.89); and A (USD$

5,789.67), marital status (widowed was not considered due to the low prevalence), work activity (no; yes), caries experience (dmft = 0; dmft ≥1), and plaque index (no participant was rated as having a poor plaque index).

To compare mean scores between subgroups, analysis of variance (ANOVA) was used and the effect size was estimated from $\eta^2_p$. The assumptions of normality and homoscedasticity were tested and confirmed for children's sex and mothers' age (skewness < 3, Kurtosis < 7, Levene's Test: $p > 0.05$). For the other variables there was heteroscedasticity (Levene's Test: $p < 0.05$), and therefore, Welch correction was performed. Multiple comparisons were performed using Tukey's or Games Howell's pos $t$-test for homo or heteroscedastic data, respectively. For the significant variables, bivariate associations were assessed using the chi-square test and the prevalence (95% CI) of caries experience and plaque index were estimated.

In addition, Pearson's correlation coefficient (r) was estimated between the dmft and the ECOHIS factors' mean scores. The level of significance adopted was 5%.

## RESULTS

A total of 371 children participated in the study (mean age 5.21 (SD = 0.64) years; 51.5% male). The mean age of mothers was 33 (SD = 7.04) years. Most children did not have caries and had a good plaque index. Most mothers were married and were from the economic strata B and C. The demographic information of the sample is shown in Table 1. The average dmft of the children was 1.36 (SD = 2.23).

Table 2 shows the descriptive statistics of the mothers' responses to the ECOHIS. None of the items showed an absolute value of skewness (<3) and kurtosis (<7), indicating adequate psychometric sensitivity of the items.

The ECOHIS model had a good fit to the sample ($\lambda = 0.65-0.88$; $\chi^2/df = 4.31$; CFI = 0.95; TLI = 0.94; RMSEA = 0.09) (Fig. 1). The convergent validity (AVE = 0.62–0.66), discriminant validity ($r_{ij}^2 = 0.59$) and reliability ($\alpha = 0.86-0.93$; $\omega = 0.80-0.80$) of the model were appropriate. The instrument also presented strong invariance between independent samples ($\Delta CFI_{M1-M0} = 0.000$; $\Delta CFI_{M2-M1} = 0.005$; $\Delta CFI_{M3-M2} = 0.004$).

Table 3 shows the mean scores for oral health impact on the child and his family according to child's sex, caries experience, and plaque index and mothers' age, marital status, work activity, and economic strata.

A significant difference in scores was found according to caries experience, plaque index and economic stratum ($p < 0.05$); children without caries and from higher income class had lower oral health impact. There was a positive and significant correlation between dmft and ECOHIS score regarding both the child ($r = 0.550$, $p < 0.001$) and the family ($r = 0.402$, $p < 0.001$).

A significant association was found between economic level and the dmft ($\chi^2 = 5.863$, $p = 0.015$) and the plaque index ($\chi^2 = 10.596$, $p < 0.001$). The prevalence of caries in children from a high economic stratum was 35.7% (95% CI [30.82–40.58]) while in those from a low economic stratum, it was 48.7% (95% CI [43.61–53.79]). The prevalence of a regular plaque index in children from the high and low economic strata was 12.1% (95% CI [8.78–15.42]) and 25.9% (95% CI [21.44–30.36]), respectively.
| Table 1 Sociodemographic characteristics of study participants. | |
|---|---|
| **Characteristic** | **n (%)** |
| **Children** | |
| **Sex** | |
| Male | 191 (51.5) |
| Female | 180 (48.5) |
| **Caries experience** | |
| dmft = 0 | 214 (57.7) |
| dmft > 0 | 157 (42.3) |
| **Plaque index** | |
| Regular | 71 (19.1) |
| Good | 300 (80.9) |
| **Mothers** | |
| **Age (years)** | |
| <30 | 123 (34.3) |
| ≥30 | 236 (65.7) |
| **Marital status** | |
| Single | 107 (29.4) |
| Married | 228 (62.6) |
| Separate | 25 (6.9) |
| Widow | 4 (1.1) |
| **Work activity** | |
| No | 124 (33.9) |
| Yes | 242 (66.1) |
| **Economic level (estimated mean family income)**[*] | |
| A (U$ 5,789.67) | 15 (4.0) |
| B (U$ 1,330.09–2,575.89) | 167 (45.1) |
| C (U$ 419.54–735.50) | 166 (44.7) |
| D-E (U$ 175.63) | 23 (6.2) |

**Notes.**
  *Values from the Brazil Economic Classification Criterion.

# DISCUSSION

The present study confirmed the validity and reliability of the data collected with the ECOHIS. The adequate fit of the original ECOHIS two-factor model to the data collaborates with findings from different contexts (*Buldur & Güvendi, 2020*; *Zaror et al., 2018*; *Randrianarivony, Ravelomanantsoa & Razanamihaja, 2020*) indicating a certain stability of the instrument.

The ECOHIS allows measuring the impact of oral problems on children and their families, which goes beyond the oral problem itself. Perception construction occurs through subjective processes and, therefore, it is a multidimensional experience (*Campos, Bonafé & Maroco, 2018*). The assessment of the mothers' perception is relevant for an integrated and comprehensive understanding of the oral health impact on children. With such information, actions can be developed considering the subject as a whole and the targeted allocation of resources (*Vieira-Andrade et al., 2015*).

**Table 2  Descriptive statistics of the ECOHIS item responses.**

| Item | Mean | Median | Standard deviation | Skewness | Kurtosis | Minimum | Maximum |
|---|---|---|---|---|---|---|---|
| it1. Has your child ever had pain in the teeth, mouth or jaws (bones of the mouth)? | 1.91 | 2 | 0.97 | 0.62 | −0.72 | 1 | 5 |
| it2. Has your child ever had difficulty drinking hot or cold drinks due to problems with teeth or dental treatments? | 1.49 | 1 | 0.80 | 1.39 | 0.62 | 1 | 4 |
| it3. Has your child ever had trouble eating certain foods due to problems with teeth or dental treatments? | 1.50 | 1 | 0.86 | 1.63 | 1.78 | 1 | 5 |
| it4. Has your child ever had difficulty pronouncing any words due to problems with teeth or dental treatments? | 1.35 | 1 | 0.79 | 2.34 | 4.87 | 1 | 5 |
| it5. Has your child ever missed daycare, kindergarten or school due to problems with teeth or dental treatments? | 1.27 | 1 | 0.66 | 2.33 | 4.39 | 1 | 4 |
| it6. Has your child ever had trouble sleeping due to problems with teeth or dental treatments? | 1.27 | 1 | 0.70 | 2.52 | 5.24 | 1 | 4 |
| it7. Has your child ever been irritated by problems with teeth or dental treatments? | 1.36 | 1 | 0.77 | 2.18 | 4.19 | 1 | 5 |
| it8. Has your child ever avoided smiling or laughing due to problems with teeth or dental treatments? | 1.34 | 1 | 0.77 | 2.55 | 6.52 | 1 | 5 |
| it9. Has your child ever avoided talking due to problems with teeth or dental treatments? | 1.33 | 1 | 0.72 | 2.49 | 6.65 | 1 | 5 |
| it10. Have you or someone else in the family ever been upset because of problems with your child's teeth or dental treatments? | 1.40 | 1 | 0.89 | 2.37 | 5.20 | 1 | 5 |
| it11. Have you or someone else in the family ever felt guilty because of problems with your child's teeth or dental treatments? | 1.55 | 1 | 1.04 | 1.82 | 2.30 | 1 | 5 |
| it12. Have you or someone else in the family missed work due to problems with your child's teeth or dental treatments? | 1.31 | 1 | 0.72 | 2.46 | 5.83 | 1 | 5 |
| it13. Has your child ever had problems with his teeth or had dental treatments that have had a financial impact on your family? | 1.32 | 1 | 0.74 | 2.50 | 5.86 | 1 | 5 |

Although the ECOHIS has been used in Brazil, (*Pahel, Rozier & Slade, 2007*; *Tesch, Oliveira & Leão, 2008*; *Nora et al., 2018*) the validity and reliability were never tested, raising questions about the quality of the evidence and consequently the conclusions obtained by the studies. In addition, few studies have applied confirmatory factor analysis for data validation (*Buldur & Güvendi, 2020*; *Zaror et al., 2018*; *Randrianarivony, Ravelomanantsoa & Razanamihaja, 2020*) a methodology strongly advised to obtain psychometric data of confirmed quality.

Moreover, no study to date has confirmed the ECOHIS invariance in independent samples to verify that the instrument's model is maintained in different samples from the same population (*Millsap & Yun-Tein, 2004*). The present study, which presented the model's strong invariance, has no similar published study to be compared.

The selection of an instrument to be used in children should consider the developmental phase of the children (*Tesch, Oliveira & Leao, 2007*). Young children have a perception of health and disease built according to their cognitive ability. According to *Rebok et al. (2001)*

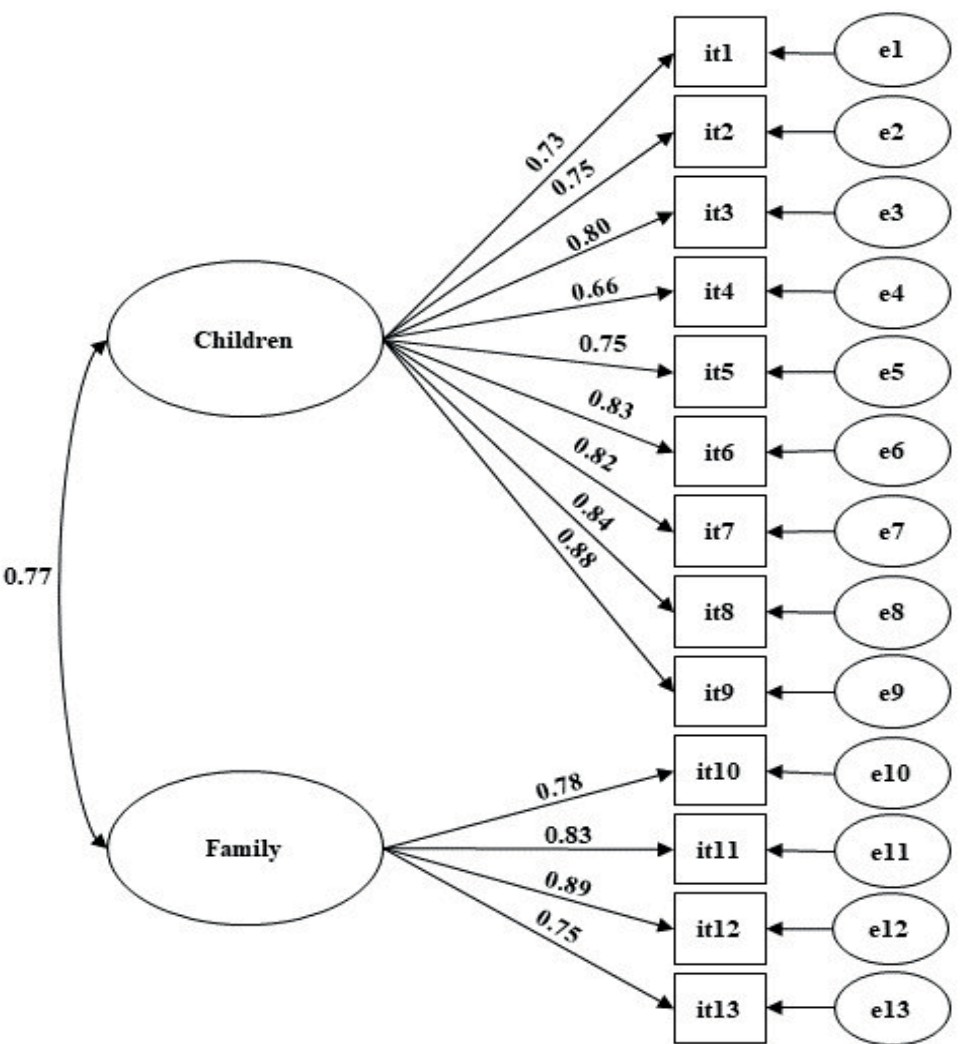

**Figure 1  Factorial model of the Early Childhood Oral Health Impact Scale (ECOHIS) adjusted for a sample of mothers of preschool children ($n = 371$).**

children under 6 years of age have difficulty in remembering events that occurred more than 24 h before, unless it is a common and essential event in their lives. Therefore, the effect of oral health events on children are better obtained through guardians.

The filling of the ECOHIS by the mothers was based on standardization requirement and on previous studies showing that women are the primary caregivers for children, although some changes in social roles and family dynamics have taken place in recent years (*Borsa & MLT, 2011*; *Wilson et al., 2014*). *Wilson et al. (2014)* highlight that the mother has a strong affective connection with her child, playing an important role in the development of habits and behaviors, including oral health promotion of children.

Children from higher income families had a lower oral health impact, which may be related to these children also being less affected by caries and having lower plaque index. These findings are consistent with studies by *Buldur & Güvendi (2020)* who found lower

**Table 3** Comparison of the mean scores of the Early Childhood Oral Health Impact Scale (ECOHIS) according to children's sex, caries experience, and plaque index and mothers' age, marital status, work activity and economic stratum.

| Characteristic | n | Mean ± SD | ANOVA | | |
|---|---|---|---|---|---|
| | | | F* | p | $\eta_p^2$ |
| **Children** | | | | | |
| **Sex** | | | | | |
| Male | 191 | 1.38 ± 0.47 | | | |
| Female | 180 | 1.44 ± 0.56 | 1.61 | 0.206 | 0.004 |
| **Caries Experience** | | | | | |
| dmft = 0 | 214 | 1.16 ± 0.28 | | | |
| dmft > 0 | 157 | 1.76 ± 0.56 | 149.50[**] | <0.001 | 0.32 |
| **Plaque index** | | | | | |
| Good | 300 | 1.30 ± 0.40 | | | |
| Regular | 71 | 1.90 ± 0.66 | 54.21[**] | <0.001 | 0.21 |
| **Mothers** | | | | | |
| **age (years)** | | | | | |
| <30 | 123 | 1.44 ± 0.53 | | | |
| ≥30 | 236 | 1.37 ± 0.49 | 1.40 | 0.238 | 0.04 |
| **Marital status** | | | | | |
| Single | 107 | 1.49 ± 0.61 | | | |
| Married | 228 | 1.38 ± 0.48 | | | |
| Separated | 25 | 1.42 ± 0.38 | 1.85[**] | 0.183 | 0.12 |
| **Work activity** | | | | | |
| No | 124 | 1.45 ± 0.60 | | | |
| Yes | 242 | 1.39 ± 0.46 | 0.96[**] | 0.328 | 0.003 |
| **Economic stratum** | | | | | |
| *C/D/E* | 189 | 1.49 ± 0.58[a] | | | |
| B | 166 | 1.34 ± 0.44[b] | | | |
| A | 16 | 1.20 ± 0.29[b] | 7.20[**] | 0.002 | 0.28 |

Notes.
*ANOVA.
[a,b,c] Different letters indicate statistical differences.
[**] Welch's F-statistic.

ECOHIS scores in children from higher income families. *Abanto et al. (2018)* reported that family income plays a protective role on the lives of preschool children. Families with more financial resources generally have better oral hygiene habits and greater access to preventive oral health care, leading to a lower oral health impact on children and their families (*Talekar et al., 2005*; *Polk, Weyant & Manz, 2010*) as found in the present study.

Although the prevalence of caries has declined in recent decades, it is still quite prevalent in low-income populations especially in 2- to 5-year-olds (*American Academy of Pediatric Dentistry (AAPD), 2014*). Our results showed that caries experience had an impact on children's lives, which corroborates finding of other studies (*Biazevic et al., 2008*; *Bekes et al., 2019*). The functional changes that accompany caries disease include difficulty in chewing and speech impairment, in addition to psychological impairment, difficulty

sleeping, and irritability (*Bönecker et al., 2012*). In addition, children's oral health impact had a positive and significant relationship with the impact on the family, as the responsibility for the children's health is generally assumed by the guardians, who often lose workdays, spend time and money on dental treatment (*Gift, Reisine & Larach, 1992*) and have to deal with a child in pain.

Children with a good plaque index showed lower ECOHIS scores, as also reported by *Bekes et al. (2019)* in a sample of German preschool children. This is directly due to the relationship between the presence of dental biofilm and the increased risk of developing oral diseases such as dental caries and periodontal diseases (*Chapple et al., 2017*).

The ECOHIS scores were not affected by children' sex and mothers' age, marital status, and works status. The results are consistent with the need to promote adequate and universal health-related preventive actions. The restricted sample used in the study may represent a limitation to the generalization of our findings to children of other ages, private schools, or their regions of the country. In addition, the cross-sectional study design does not allow cause and effect inferences.

This study presents information relevant to health professionals by providing the validity and reliability estimates of data obtained with the ECOHIS and by exploring the impact of oral problems on preschool children and their families. The findings can guide the development of comprehensive educational and preventive actions and treatment strategies, underscoring the need to prioritize public health programs in economically vulnerable groups.

These results may be important both for the development of future research protocols and for directing clinical interventions that use the investigated variables, opening the possibility of a more integral and comprehensive care of dental patients in a way that is centered on the patients' well-being, which will certainly enrich the decision-making process, and may also improve the individual's adherence to treatment and their awareness of their health.

## CONCLUSIONS

The data obtained with the ECOHIS from mothers of preschool children were valid, reliable, and invariant between independent samples. The economic stratum, the caries experience, and the level of plaque had a significant impact on children and their families.

### Funding

This study was supported by the Coordenação de Aperfeiçoamento de Pessoal de Nível Superior–Brasil (CAPES)–grant Code 001 and the São Paulo Research Foundation (Fapesp) (Proceedings # 2019/17200-9 and # 2019/24424-0). The funders had no role in study design, data collection and analysis, decision to publish, or preparation of the manuscript.

## Grant Disclosures

The following grant information was disclosed by the authors:

Coordenação de Aperfeiçoamento de Pessoal de Nível Superior–Brasil (CAPES): Code 001.

São Paulo Research Foundation (Fapesp): 2019/17200-9, 2019/24424-0.

## Competing Interests

The authors declare there are no competing interests.

## Author Contributions

- Bianca Núbia Souza Silva conceived and designed the experiments, performed the experiments, analyzed the data, prepared figures and/or tables, authored or reviewed drafts of the article, and approved the final draft.
- Lucas A. Campos conceived and designed the experiments, performed the experiments, analyzed the data, prepared figures and/or tables, authored or reviewed drafts of the article, and approved the final draft.
- João Marôco conceived and designed the experiments, analyzed the data, authored or reviewed drafts of the article, and approved the final draft.
- Juliana A.D.B Campos conceived and designed the experiments, analyzed the data, prepared figures and/or tables, authored or reviewed drafts of the article, and approved the final draft.

## Human Ethics

The following information was supplied relating to ethical approvals (i.e., approving body and any reference numbers):

Research Ethics Committee of the Faculty of Pharmaceutical Sciences of Araraquara (UNESP)

## Data Availability

The raw data are available in the Supplemental Files.

## Supplemental Information

Supplemental information for this article can be found online at http://dx.doi.org/10.7717/peerj.16035#supplemental-information.

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
