# Peer review of "The Early Childhood Oral Health Impact Scale (ECOHIS): psychometric properties and application on preschoolers"

_PeerJ, doi:10.7717/peerj.16035_

## Round 0.1 · original submission · Minor Revisions

Dear Dr Campos,

Thank you for your submission to PeerJ; this is a good article with a nice experimental design, and the article's content is within the scope of PeerJ.

Reviewer 1 ·

Basic reporting

The manuscript was well written in English using clear, unambiguous, technically correct text and conformed to professional standards expression.
There is sufficient introduction and background to demonstrate how the work fits into the broader field of knowledge with relevant and appropriately referenced literature.
The structure of the manuscript basically conforms to the journal’s Instructions for Authors.
Figures are appropriately described and labelled.
The submission includes all results relevant to the aims and objectives of the study.

Experimental design

The research question was well defined, relevant & meaningful and it also identified a knowledge gap.
The study contributes towards targeted clinical intervention.
It is clear that the study followed the necessary ethical standards.
(i). However, appreciate the authors explain why ethical approval was provided by the Research Ethics Committee of the Faculty of Pharmaceutical Sciences of Araraquara instead of Faculty of Dentistry or of Oral Diseases. As the study is entirely dental based. (P6, L78)

The study methods are generally described with sufficient detail & with enough information to be reproduced.
(ii). It would however be interesting to know a little more about the subjects in this study. Why were they chosen from the Child Education and Recreation Centers (CER) of Araraquara? Is its population representative of all preschool children aged between 4 to 6 years in Brazil?
(iii). The authors also mentioned that dental caries was diagnosed based on the World Health Organization (WHO) criteria using the ceod index. Please highlight on the coed index and its background.

Validity of the findings

It cannot be said that the demographic and oral characteristic findings of this study are of high impact or novelty, but they demonstrate high health intervention implications, therefore it adds value to the literature.
All underlying data have been provided, are robust and statistically sound.
The data on which the conclusions are based are made available in an acceptable repository.
Conclusions are well stated and linked to the original research question and limited to the supporting results.

Additional comments

No further comments

Reviewer 2 ·

Basic reporting

No comment

Experimental design

This is a good article with a nice experimental design.
The methodology states that the study used only natural light and wooden spatula- Whereas caries detection is done either by using explorer or X rays_ please clarify on this.

Validity of the findings

The study is valid and they examined the subjects with a week time gap- to eliminate the bias

Additional comments

Overall, it's a good article. It gives good insight into the impact of socio-economic status on the preschool children on their oral hygiene habits and their output.

·

Basic reporting

Thanks for this interesting article. It is written in clear, unambiguous and professional English. The introduction sets out the context and the references are properly referenced and relevant. The tables and figure support the content, and the underlying data has been provided.

I have the following minor comments:
1) There is a small typo in line 101 (“siting”) and in line 81 “studies” should be capitalized as part of the proper noun STROBE.
2) In the abstract, the result section is included in the methods section. I suggest moving the results to a separate paragraph.
3) In table 2 the spelling is divergent: “ceod” vs “Ceod”. I propose to use equal spelling.
4) Table 3 includes superscripts “a”, “b” and “c” and double asterisk. I propose adding explanations below the table as footnotes.

Experimental design

The article's content is within the scope of PeerJ. The research question addresses the reported knowledge gap. The methods are described with sufficient detail.

Validity of the findings

Most of the calculations can be replicated and support the conclusion, but the following issue came up:

5) Analyzing the raw data, I came to the conclusion that 213 subjects have a ceod of zero and 158 subject have a ceod bigger than zero. This is different from the numbers reported in the article and table 1.

I can not comment on the economic classification and the Confirmatory Factor Analysis.

Reviewer 4 ·

Basic reporting

No comments

Experimental design

Well defined with appropriate test used

Validity of the findings

No comments

Additional comments

1. Title - Since the study aimed to assess the fitness of ECOHIS, the title didnt reflect the study. Author may add 'fitness of ECOHIS...sensitivity of ECOHIS etc2..' into the title to represent the study
2. Line 93 - Was the questionnaire given to mothers are in English or translated?
3. Line 101 - exams should be fully written as examination

·

Basic reporting

The article is written in clear and well understandable English that is easy to be read. Sufficient literature references (47).
Some technical mistakes:
On line 53 - empty space in the beginning of the line.
Line 58 - "is" included.
Line 190 - collaborates

Experimental design

The present study has no similar published study and this makes it valuable.
I would recommend to add few more sentences regarding table1 and 2, and figure 1 so to make it more clear for understanding for the readers.
Also to add some more information regarding ECOHIS and plaque index for those readers who are not dentists or psychologists.
The study is really of great interest and these minor correction will increase its value for a wider range of readers.

Validity of the findings

All data have been provided and statistically proven.

Additional comments

It was very interesting to be to read this easy-to-read manuscript and I am sure that will be of great value for the journal.

---

## Round 0.2 · accepted · Accept

Dear Authors

in agreement with the reviewers I believe that your manuscript can be accepted for publication.

Congratulations!

Reviewer 2 ·

Basic reporting

Made all the needed corrections

Experimental design

good

Validity of the findings

valid and significant

Additional comments

Good

·

Basic reporting

Thank you for submitting the revised article. I want to make only two minor comments regarding basic reporting:

1) Please correct the spelling error "wiil" to "will" line 120.

2) In line 158 the second group of the "age of mothers" feature should be corrected to greater than or equal to 30 years old.

Experimental design

No comments regarding experimental design.

Validity of the findings

I have no new comments regarding validity of findings.

Reviewer 4 ·

Basic reporting

Clear and concise

Experimental design

Appropriate with study design

Validity of the findings

No comments

Additional comments

Very minor comments:

Line 118 - '..wiil..' should be '..wil..'
Line 118 - ..the following order tooth 55-51..(add tooth before numbering)

·

Basic reporting

Clear and professional English.

Experimental design

Methods are described with sufficient details.

Validity of the findings

Sufficient literature review.

Additional comments

I would suggest in Table 1 and Table 3 to write the same characteristic- "Work activity" or " Work".
Also, on line 118 there is a technical mistake - the word "will" is written as " wiil".
Thank you!